# Facile Synthesis of Mesoporous Silica at Room Temperature for CO_2_ Adsorption

**DOI:** 10.3390/mi13060926

**Published:** 2022-06-10

**Authors:** Misun Kang, Jong-tak Lee, Min-Kyoung Kim, Myunghwan Byun, Jae-Young Bae

**Affiliations:** 1Department of Advanced Materials Engineering, Keimyung University, Daegu 42601, Korea; misun.kang@gmail.com; 2Department of Chemistry, Keimyung University, Daegu 42601, Korea; atomos27@daum.net (J.-t.L.); gyeong0730@naver.com (M.-K.K.)

**Keywords:** mesoporous silica, room-temperature synthesis, CO_2_ adsorption, high pore volume

## Abstract

Although mesoporous silica materials have been widely investigated for many applications, most silica materials are made by calcination processes. We successfully developed a convenient method to synthesize mesoporous materials at room temperature. Although the silica materials made by the two different methods, which are the calcination process and the room-temperature process, have similar specific surface areas, the silica materials produced with the room-temperature process have a significantly larger pore volume. This larger pore volume has the potential to attach to functional groups that can be applied to various industrial fields such as CO_2_ adsorption. This mesoporous silica with a larger pore volume was analyzed by TEM, FT-IR, low angle X-ray diffraction, N_2_-adsorption analysis, and CO_2_ adsorption experiments in comparison with the mesoporous silica synthesized with the traditional calcination method.

## 1. Introduction

Global energy production systems need to burn fuels, such as petroleum, coal, and various natural gases, to support electricity, heat, and other energy consumers [1]. However, heavy uses of fossil fuels generate large amounts of harmful pollutants and gases such as carbon dioxide (CO_2_), carbon monoxide (CO), sulfur dioxide (SO_2_), and nitrogen oxides (NO_2_) that can decrease the air quality and affect the climate. In particular, CO_2_ is a representative gas of global warming with many environmental issues. Thus, the adsorption of CO_2_ gases is one of the most important research topics for solving severe climate crisis [2,3,4].

Carbon capture and sequestration (CCS) is known to be one of the most effective ways to lower greenhouse gas emissions [5]. The basic concept of CCS is to capture CO_2_ molecules generated from industrial processes and transport them to storage [6]. As an alternative to CCS, post-combustion capture (PCC) is frequently used. In particular, PCC using a sorption-based process is quite promising since it has high efficiency and selectivity [7,8].

Nowadays, CO_2_ adsorption studies based on amine functional groups have been intensively and extensively performed. Various materials with a high specific surface area such as zeolite [9,10,11], activated carbon [12,13,14,15], metal–organic frameworks (MOFs) [16,17,18,19], and porous silica [20,21,22,23] have been demonstrated. When an amine functional group meets carbon dioxide gas in the presence of moisture, a carbamate salt and a hydrogen carbonate salt are formed. Thus, an amine functional group is known to adsorb CO_2_ with moisture stability [24]. Although many materials with an amine functional group have already been introduced, porous silica materials with a high specific surface area and high thermal stability account for the most.

Porous silica with an excellent specific surface area can be produced via a self-assembly method using surfactants. When the surfactant reaches a certain concentration, spherical micelles are formed, which is the so-called “critical micelle concentration 1 (CMC1)”. After slightly increasing the concentration, spherical micelles will grow as rods. This concentration is called “critical micelle concentration 2 (CMC2)”. According to the arrangement of rod-shaped micelles, pore structures can be divided into hexagonal, cubic, and lamellar structures. Porous silica synthesized using a surfactant has the advantage of supporting many functional groups due to its high specific surface area and pore volume [25,26]. However, to produce a porous silica, a calcination treatment is needed to remove the surfactant that forms in the pore structure. In general, the cost of this calcination process is quite high because the surfactant can only be removed via a high-temperature treatment at 400 °C or higher [27,28].

In this study, a method for synthesizing a porous silica with a high specific surface area at room temperature without needing any high-temperature heating treatments was developed. In addition, the porous silica prepared with this room-temperature synthesis method was confirmed to have an excellent pore volume with a high CO_2_ adsorption performance as an amine support. It can open a brand-new path for developing the next generation of CO_2_ adsorption technology.

## 2. Materials and Methods

### 2.1. Materials

For the synthesis of porous silica, sodium silicate solution (28–30 wt%, DAEJUNG, Gyeonggi-do, Korea) and cetyltrimethylammonium chloride (CTACl, 25 wt%, Sigma-Aldrich-Korea, Seoul, Korea) were used as a silica precursor and a surfactant, respectively. Hydrochloric acid (HCl, 35–37%, Samchun, Gyeonggi-do, Korea) was utilized as a pH control. Absolute ethyl alcohol (99.9%, DAEJUNG, Gyeonggi-do, Korea) was employed as a solvent during the surfactant removal process in which the synthesized silica was washed at room temperature several times. Finally, amine functional groups were introduced into the synthesized mesoporous silica using tetraethylene pentamine (TEPA, 93%, Kanto Chemical, Tokyo, Japan).

### 2.2. Porous Silica Synthesis Using the Calcination Process

For comparison with the porous silica synthesized at room temperature, porous silica was synthesized with the conventional method involving a calcination process. In a typical synthesis, 40 mL of sodium silicate solution was added to 700 mL of distilled water. The resulting solution was stirred at room temperature for 0.5 h, followed by the addition of 20 mL of CTACl and was stirred at room temperature for another 24 h. The above solution was then centrifuged at 10,000 rpm for 5 min to obtain a white slurry. The resulting slurry was dried at 80 °C for 8 h in a convection oven. The dried material was finely ground in a mortar and calcined at 600 °C for 5 h to produce a porous silica (T-MS-0).

### 2.3. Porous Silica Synthesis Using the Room-Temperature Process

The process to synthesize porous silica through a room-temperature process proceeded in a similar way to the first few steps mentioned above. First, 40 mL of sodium silicate solution was added to 700 mL of distilled water and stirred at room temperature for 0.5 h. Next, 20 mL of CTACl was introduced into the above solution and stirred at room temperature for 0.5 h. The pH of the resulting solution was adjusted to 10 by dropwise addition of 5.0 M aqueous hydrochloric acid solution and stirred for another 24 h. After finishing the stirring process, the solution was centrifuged at 10,000 rpm for 5 min to obtain a white slurry. Since the obtained substance contained surfactant, an ethanol solution was prepared to remove the surfactant. After preparing 1 L of hydrochloric acid/ethanol solution having a concentration of 1.0 M, the slurry obtained in the previous process was added to the removal solution and stirred for 1~2 days. The final solution was centrifuged at 10,000 rpm for 5 min. The resulting slurry was washed twice with an anhydrous ethanol solvent. The washed material was dried in a convection oven at 60 °C for 8 h. The dried material was finely ground using a mortar to obtain a porous silica (R-MS-0).

### 2.4. Introduction of Amine Group

Two types of synthesized porous silica (T-MS-0 and R-MS-0) were used as amine supports for CO_2_ adsorption. Each 0.5 g of synthesized T-MS-0 and R-MS-0 was added to 50 mL of absolute ethanol, respectively. Prepared materials were dispersed by stirring at room temperature for 0.5 h. After adding 1 g of TEPA into each dispersed solution, the resulting material was stirred at room temperature for 1 h. Each solvent was removed from the solution after stirring at 50 °C for 20 min using a rotary evaporator. After removing the solvent, each sample was dried in a convection oven at 100 °C for 1 h to obtain T-MS-N and R-MS-N into which amine groups were introduced.

### 2.5. Characterization

Pore morphology of porous silica was analyzed using a field emission transmission electron microscope (FE-TEM; HF-3300) at an operating voltage of 300 kV. N_2_-adsorption-desorption isotherm (N_2_-sorption; QUANTACHROME, Qudrasorb SI) was used to determine specific surface area and pore distribution of the porous silica with or without a calcination process. The measured temperature was maintained at 77 K using liquid nitrogen. The adsorbed nitrogen was normalized to standard temperature and pressure. Prior to analysis, heat treatment was performed at 200 °C for 6 h to remove moisture and impurities adsorbed on the surface of the sample. The Brunauer–Emmett–Teller (BET) specific surface area was calculated from the linear part (P/P_0_ = (0.05–0.30) of the BET equation. The volume and size of pores were calculated using the Barrett–Joyner–Halenda (BJH) equation. X-ray diffractometer (XRD; PANalytical X’pert PRO MRD) was used to analyze the pore structure of the porous silica with or without a calcination process. Measurements were performed in 2θ scan mode with Cu-Kα rays (λ = 0.0154 nm). A Fourier transform infrared spectrometer (FT-IR; thermo, Nicolet iS50) was used to determine whether the surfactant in the powder of porous silica was removed through a room-temperature process. An energy-dispersive X-ray spectrometer (EDS; JEOL, EX-746OOU4L2Q) was used for elemental distribution analysis of amine-group-introduced porous silica. The working distance (WD) was set to be 9.5~10.5 mm and counts per second (CPS) was set at 300,000 or more during the entire analysis to ensure reliability. In order to confirm the gas adsorption capacity of the amine-group-introduced porous silica, carbon dioxide adsorption performance was carried out using a gas chromatography (GC; HP 6890) system equipped with a thermal conductivity detector (TCD). While analyzing the CO_2_ adsorption performance, a nitrogen-based mixed gas having a carbon dioxide concentration of 30% was utilized. The gas flow rate was set at 5 mL/min using a mass flow controller (MFC). High-purity helium gas was used as the makeup gas.

## 3. Results and Discussion

Since the traditional synthesis of mesoporous silica via calcination applies heat above 600 °C, the surfactant in the sample is naturally removed. On the other hand, the room temperature synthesis requires an elimination process for the surfactant. Therefore, before discussing the results of other experiments, an important point is to confirm that the surfactant was eliminated from the room-temperature synthesis of mesoporous silica. Figure 1 shows the FT-IR spectra for confirming the removal of the surfactant, which was CTACl used for the room-temperature synthesis of R-MS-0. Peaks at 2800–3000 cm^−1^ and around 1500 cm^−1^ of the magenta line in Figure 1 corresponded to the bands for the C–H stretching vibration and the C–H scissoring in CTACl, respectively. The peak at 1220 cm^−1^ was also assigned to the stretching vibration for C–N in CTACl [29]. However, after an elimination process of the surfactant for 24 or 48 h, it was verified that the surfactant of the room-temperature-synthesized mesoporous silica was removed based on the disappearance of the peaks related to CTACl such as the C–H and C–N bonds. For subsequent experiments, the surfactant elimination process was performed for 24 h. The black line spectrum showed T-MS-0, which are the mesoporous silica materials, after the calcination process to remove the surfactant.

According to the TEM images shown in Figure 2a,b, the porous silica (T-MS-0) is synthesized in the traditional way including a calcination process consisting of uniformly arranged pores. This was because the thermal energy during the calcination allowed for uniform and well-ordered pores. On the other hand, R-MS-0 (the porous silica synthesized with the room-temperature method) consisted of irregular and unordered pores as shown in Figure 2c,d.

The TEM images corresponded to the spectra of low-angle X-ray diffraction as shown in Figure 3. The T-MS-0 with the uniform and well-ordered pores was confirmed by a sharp peak at around 2.5 degrees in 2θ. Compared to T-MS-0, the low-angle XRD pattern of R-MS-0 showed a wide peak in 2θ from 1.1 to 2.1 degrees and a left-shifted peak, indicating various and larger pore sizes than T-MS-0. As mentioned earlier, the spectral difference in Figure 3 was caused by the heat treatment. The thermal energy given during the calcination process makes pores consistent and orderly.

The specific surface areas and pore volumes of T-MS-0 and R-MS-0 were investigated using nitrogen (N_2_) adsorption–desorption isotherm graphs as shown in Figure 4. Both T-MS-0 and R-MS-0 exhibited a characteristic type IV BET hysteresis loop, meaning that sizes of porous pores were in the range of 1.5 to 100 nm. BET-specific surface areas were deduced to be 1139.911 m^2^/g and 1072.455 m^2^/g, respectively. In Figure 4, the two graphs showed similar adsorption curves within a relative pressure from 0 to 0.3, indicating that the specific surface areas of these two silica materials were similar. However, when the relative pressure was more than 0.45, the isotherm graphs of R-MS-0 and T-MS-0 showed different tendencies. The adsorption–desorption curves of R-MS-0 were largely separated. The volume adsorption at a relative pressure of 1.0 for R-MS-0 was also higher than that for T-MS-0. These results imply that the amounts of adsorbed gases by R-MS-0 and T-MS-0 were different as the pore volumes of these two silica materials were different.

Since the average pore size was derived from the pore volume that was estimated from the amount of adsorbed gas in the N_2_ adsorption–desorption isotherm curve, the specific pore sizes of T-MS-0 and R-MS-0 were denoted as shown in Figure 5. The overall results, including surface area, pore volume, and average pore size, are shown in Table 1. Although the average pore sizes of the two samples were similar, the pore volume of R-MS-0 was twice that of T-MS-0. Irregular and jumbled pores in R-MS-0 were large in places (shown in the TEM images in Figure 2), which could explain why R-MS-0 pores had twice the volume of T-MS-0 pores. In addition, a larger pore volume of R-MS-0 could provide support for amine functional groups inside these pores for CO_2_ adsorption.

To attach amine functional groups into R-MS-0 and T-MS-0, TEPA was chosen (see the Materials and Methods section). R-MS-0 and T-MS-0 loaded with amine functional groups were determined by EDS several times to check their amounts of amine functional groups. The EDS results and mass differences after introducing the amine groups into the silica materials indicated that both silica materials had similar amounts of amine groups attached. The FT-IR spectra shown in Appendix A also indicated that the amine functional group was attached to the silica materials due to the peaks between 3500 and 3300 cm^−1^. The CO_2_ adsorption performances of these amine-group-loaded silica materials were then tested at various intervals. As shown in Figure 6, although the pore size and average quantity of the loaded amine group of R-MS-N were similar to those of T-MS-N, the amount of CO_2_ adsorption of R-MS-N was more than twice that of T-MS-N. Such a difference in the amount of CO_2_ adsorption between these two silica materials was due to the differences in their pore volumes caused by the non-uniformly and irregularly generated pores of R-MS-0. Because R-MS-0 did not have a calcination process that gave thermal energy, pores of various sizes including large ones allowed for a significantly higher volume of R-MS-0. Table 2 shows the pore volumes, quantity differences of silica materials after loading with amine groups, EDS data, and the amount of CO_2_ adsorption.

Figure 7 describes how the large pore volume of R-MS-N affects CO_2_ adsorption. Figure 7a shows the white powder of T-MS-0 including a calcination step and the white powder of R-MS-0 prepared at room temperature. Figure 7b denotes powder images of T-MS-N and R-MS-N after attaching amine functional groups to these silica materials using TEPA. Unlike the white color of R-MS-N, the color of T-MS-N is yellowish, although both types of silica materials have almost the same amount of amine groups. The yellow color originated from the relatively large amount of amine groups, which are attached outside of the porous silica materials [30]. For the inside of both silica materials, a greater number of amine groups are supported inside the pores of R-MS-N due to its pore volume being larger than T-MS-N. However, for the outside of the pores, more quantities of amine groups are attached to the outside of the pores of T-MS-N than those of R-MS-N. According to a previous study [31], supported amine groups inside the silica pores can enhance the CO_2_ adsorption performance compared to the supported ones outside the silica pores. As a result, R-MS-N has an improved CO_2_ adsorption ability because of its larger pore volume.

## 4. Conclusions

In this study, a method for synthesizing a porous silica (R-MS-0) with excellent pore volume through a simple room-temperature process is successfully developed. Although the porous silica does not have a regular pore structure, the large pore volume capable of supporting the functional groups of carbon dioxide adsorption is obtained. The porous silica synthesized at room temperature has a value of specific surface area similar to that of the porous silica (T-MS-0) synthesized using the traditional method. However, the CO_2_ adsorption performance of R-MS-N is increased by 2.1 times that of T-MS-N due to the larger pore volume of R-MS-0. The large specific surface area and excellent pore volume of the porous silica prepared by a simple room-temperature process propose a great potential for CO_2_ adsorption as well as studies of other catalysts or adsorption supports.

## Figures and Tables

**Figure 1 micromachines-13-00926-f001:**
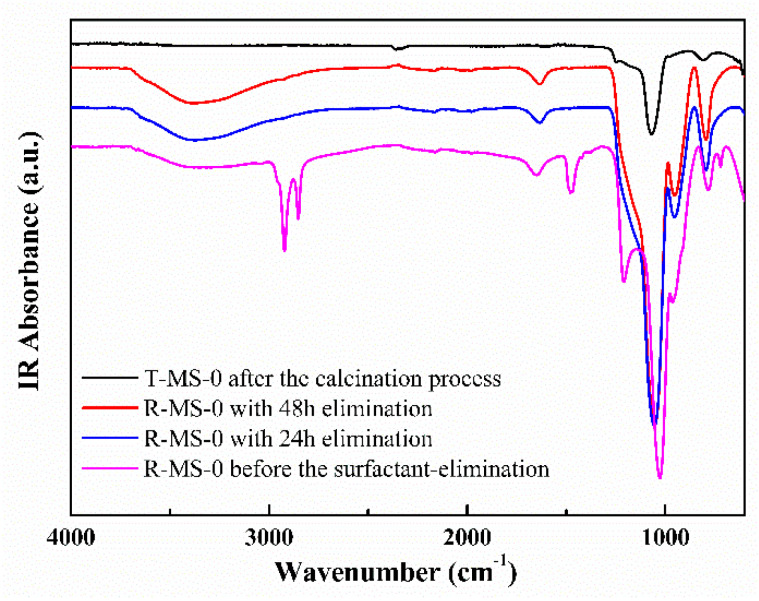
FT-IR spectra of the room-temperature-synthesized mesoporous silica. The magenta line represents the spectrum before the removal process of the surfactant, and the red and blue lines indicate the spectrum after the elimination processes for 48 and 24 h, respectively. The black line is the FT-IR spectrum of mesoporous silica made by the calcination process.

**Figure 2 micromachines-13-00926-f002:**
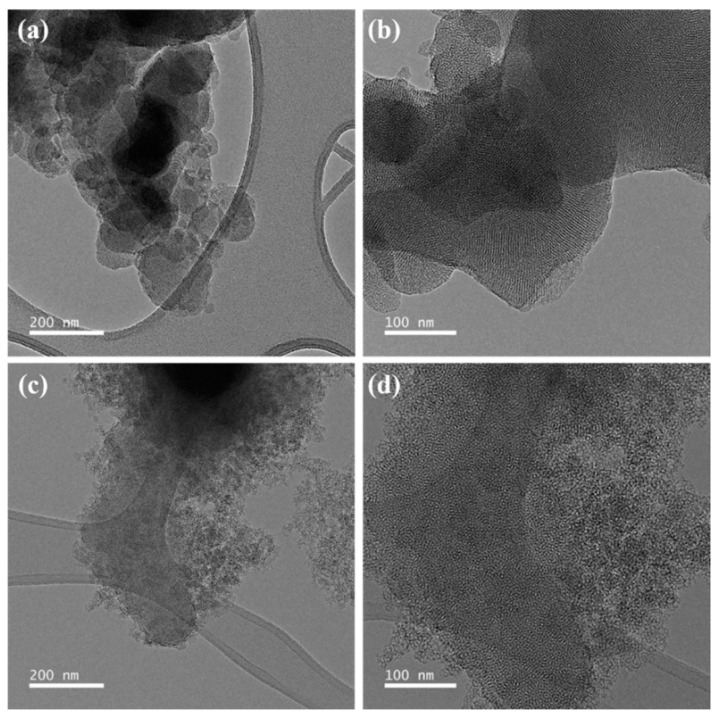
TEM images of the T-MS-0 in (**a**,**b**) and the R-MS-0 in (**c**,**d**).

**Figure 3 micromachines-13-00926-f003:**
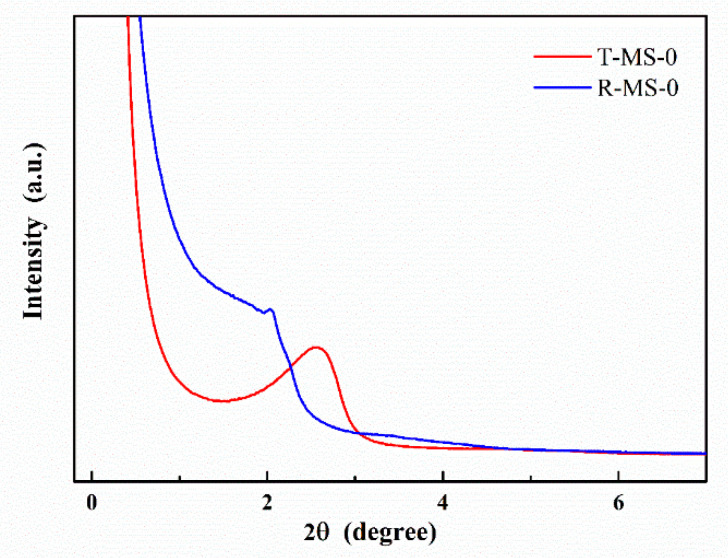
Low-angle XRD spectra of the T-MS-0 and R-MS-0.

**Figure 4 micromachines-13-00926-f004:**
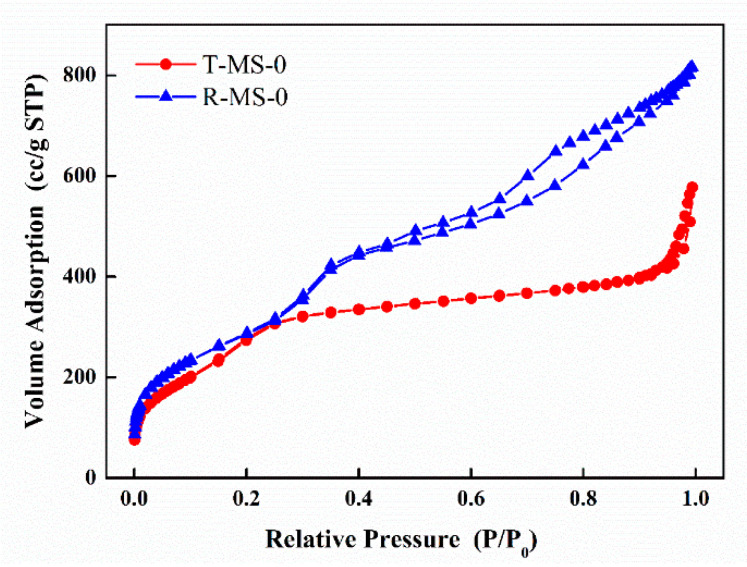
The nitrogen (N_2_) adsorption–desorption isotherm curves of T-MS-0 and R-MS-0.

**Figure 5 micromachines-13-00926-f005:**
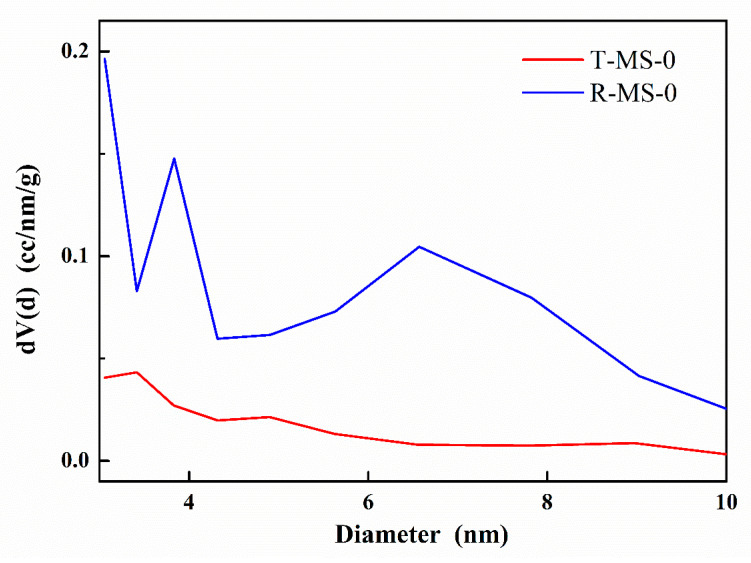
Specific pore size distributions for T-MS-0 and R-MS-0.

**Figure 6 micromachines-13-00926-f006:**
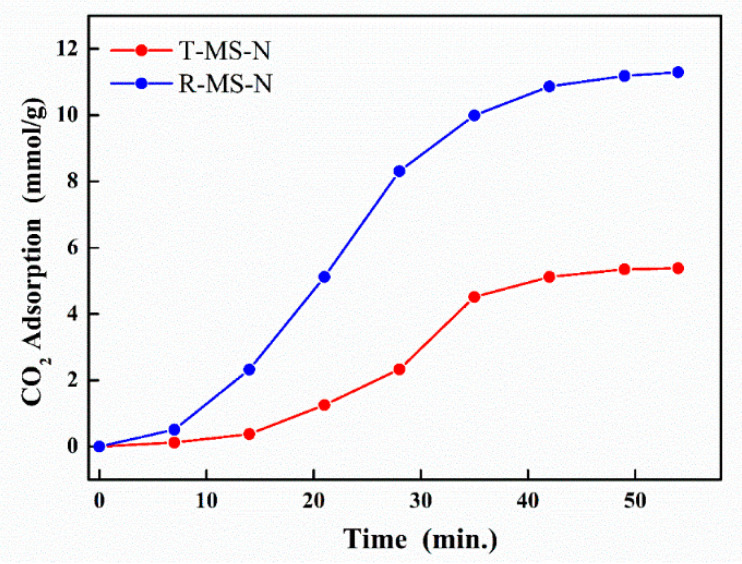
CO_2_ adsorption performance of T-MS-N and R-MS-N.

**Figure 7 micromachines-13-00926-f007:**
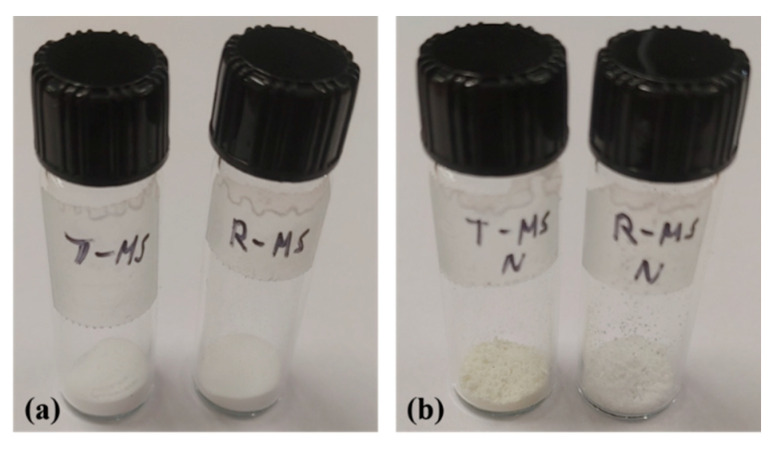
Images of porous silica (**a**) before the process for introducing the amine group (T-MS-0 and R-MS-0) and (**b**) after the process for introducing the amine group (T-MS-N and R-MS-N).

**Table 1 micromachines-13-00926-t001:** The textural properties of T-MS and R-MS.

Sample Name	Surface Area (m^2^/g)	Pore Volume (cm^3^/g)	Average Pore Size ^1^ (nm)
T-MS-0	1139.911	0.443	3.417
R-MS-0	1072.455	0.845	3.058

Note: ^1^ Diameter of window, determined from desorption branch according to BJH method.

**Table 2 micromachines-13-00926-t002:** Pore volume, EDS, and CO_2_ adsorption results of T-MS-N and R-MS-N.

Sample Name	Pore Volume (cm^3^/g)	EDS Data				CO_2_ Adsorption (mmol/g)	Mass Variation after Introducing Amine Functional Groups (g)	
		Atom Si (%)	Atom N (%)	Atom C (%)	Atom O (%)		Before	After
T-MS-N	0.443	6.66	18.32	49.85	25.18	5.384	0.51	1.07
							0.56	
R-MS-N	0.845	8.65	22.98	48.06	20.31	11.296	0.49	1.07
							0.58	

## Data Availability

Not applicable.

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
