# Peer review of "Facile Synthesis of Mesoporous Silica at Room Temperature for CO2 Adsorption"

_micromachines, 2022, doi:10.3390/mi13060926_

Round 1

Reviewer 1 Report

The comments are included in the attached PDF

Reviewer 2 Report

The paper titled "Facile Synthesis of Mesoporous Silica at Room Temperature for CO2 Adsorption", reports on silica materials produced at room temperature with large pore volume suitable for CO2 adsorption with the aid of amine support.

The paper has sufficient scientific value and originality in its technical content to merit publication. The paper contains good structural and textural characterization, and the text itself is easy to follow and understand.

The paper is generally acceptable for publication after the minor corrections (comments are in red):

• Page 2, line 91: “The pH of the resulting solution was adjusted to 10…” Authors should explain why the pH is adjusted.

• Page 4, line 166: instead of Figure 1. TEM images of …, should stand Figure 2. TEM images of…

• Page 5, line 175: instead of Figure 2. Low angle…, should stand Figure 3. Low angle…

• Page 5, line 190: instead of Figure 3. The nitrogen…, should stand Figure 4. The nitrogen…

• Page 6, line 204: instead of Figure 4. Specific pore size…, should stand Figure 5. Specific pore size…

• Page 7, lines 209-221: To attach amine functional groups into R-MS-0 and T-MS-0, TEPA is chosen (see the experimental section). Each R-MS-0 and T-MS-0 loaded the amine functional groups was determined by EDS several times to check the amount of amine functional groups. The EDS results and the mass differences after introducing the amine groups process into the silica materials indicate that both silica materials have similar amounts of amine groups attached. Then, with the amine group loaded silica materials, the CO2 adsorption performance is tested depending on the times.

To attach amine functional groups into R-MS-0 and T-MS-0, TEPA was chosen (see the experimental section). R-MS-0 and T-MS-0 loaded with amine functional groups were determined by EDS several times to check their amounts of amine functional groups. EDS results and mass differences after introducing amine groups into silica materials indicated that both silica materials had similar amounts of amine groups attached. CO2 adsorption performances of these amine group loaded silica materials according to time were then tested.

Comment: The highlighted text is doubled, and it should be deleted.

• Page 7, line 233: instead of Figure 5. CO2 adsorption…, should stand Figure 6. CO2 adsorption…

• Page 8, line 253: instead of Figure 6. Images of…, should stand Figure 7. Images of…

Round 2

Reviewer 1 Report

I accept the current version of the manuscript. Tahnk you for the explanations